# HMV-CL: Heterogeneous Multi-View Contrastive Learning for Improved Representation of Rare Disease Patient Narratives

## Abstract

Narratives from rare disease patients and families on social media offer valuable insights for public health researchers. However, pretrained text embeddings present limitations that are exacerbated by these very specific High Dimensional - Low Sample Size data, resulting in poor downstream performance. We propose a heterogeneous multi-view contrastive learning framework that aligns dense `CamemBERT-v2` embeddings with sparse symbolic views (clinical signs, medical procedures and pharmacological treatments) extracted from the posts. Our method considerably improves embedding space geometry by resolving embedding anisotropy (reducing mean cosine similarity from 0.88 to 0.10) and increasing space utilization (effective rank rising from 27% to 70%); it also improves downstream clusterability, achieving a stable Adjusted Rand Index of 0.71 and a drastic reduction in the outliers ratio from 95.15% to 10.59%. Moreover, low inter-view correlations ($< 0.5$) provide natural regularization, preventing collapse without explicit penalties. Overall, we established a strong baseline for patient-centered NLP in the underserved domain of rare diseases, that will hopefully lead to more work on the subject, including benchmark datasets, formal convergence guarantees, and concrete analyses of rare disease patients' trajectories and needs.

## 1. Introduction

Self-supervised learning (SSL) has become a central approach for learning rich representations from large collections of unlabeled text, enabling strong downstream performance. SSL-based language models such as Bidirec-

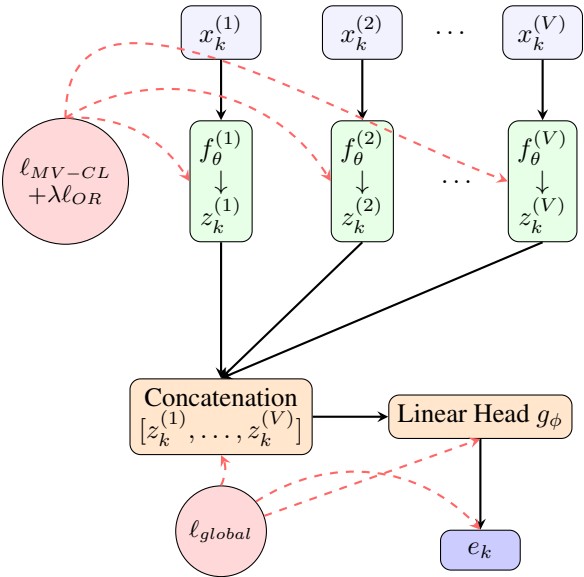

*Figure 1.* Overview of the HMV-CL framework. Heterogeneous views $x_k^{(v)}$ are projected into a shared latent space through view-specific encoders $f_\theta^{(v)}$. The final document embedding $e_k$ is generated

tional Encoder Representations from Transformers (BERT) (Vaswani et al., 2017; Devlin et al., 2019; Gao et al., 2019) provide generic embeddings that can be adapted to many applications, including clinical and biomedical Natural Language Processing (NLP). However, in practice, it is still a mathematical and computational challenge to extract relevant document representations from these models, especially when the data distribution significantly differs from the pretraining corpora (Ethayarajh, 2019; Li et al., 2020).

A well-documented issue is embedding anisotropy: Gao et al. (2019) showed that most of learned word embeddings tend to be distributed in a narrow cone. Ethayarajh (2019); Li et al. (2020) showed that pretrained Transformer embeddings and sentence embeddings tend to also suffer from this phenomenon and to concentrate in a few dominant directions, reducing discriminative power, harming clustering

[1]Anonymous Institution, Anonymous City, Anonymous Region, Anonymous Country. Correspondence to: Anonymous Author <anon.email@domain.com>.

Preliminary work. Under review by the International Conference on Machine Learning (ICML). Do not distribute.

and retrieval and more generally complicating downstream analyses.

In highly specialized domains, these limitations related to the structure of the embedding space are exacerbated. In health-related social media, and especially in rare disease communities, the language used by patients and caregivers differs greatly from the corpora used to train domain-specific models such as `CamemBERT-bio` (Touchent et al., 2023) or `DrBERT` (Labrak et al., 2023)[1], making them even less adapted than general models such as `CamemBERT` (Martin et al., 2020; Antoun et al., 2024; Le Priol et al., 2024). This happens because these narratives are highly heterogeneous, mixing highly specialized and potentially rare medical vocabulary and informal and experiential discourses.

For many of these rare conditions, patients' or caregivers' narratives are scarce. But, in the same time, social media and specialized forums have become crucial spaces for families to share their experiences, exchange knowledge and discuss treatments (Golder et al., 2024), making these narratives a novel unique and valuable source of real-world evidence about lived experiences and natural histories, that could enrich existing clinical and scientific data. But their analysis requires adapted representation learning methods, able to cope with their specifities.

**Research questions** Our work aims at answering the following research question: how can we produce less anisotropic, more informative and clusterable document embeddings for rare disease social media data?

This work also answers to a secondary research question: can the heterogeneous nature of the views used alleviate the potential redundancy between views and consequent dimensional collapse?

**Contributions** We propose a multi-view contrastive framework for learning document-level embeddings from French rare disease social media data. While traditional NLP representation learning methods often focus exclusively on text, we wanted to leverage, alongside standard text embeddings, complementary views extracted from patient narratives: clinical signs, medical procedures, and pharmacological treatments.

Our method optimizes a multi-view contrastive objective that aligns representations of the same document across views. We also found that the redundancy between views is not an issue with our specific views, which are heterogeneous enough to act as natural regularizer and thus avoid representational collapse.

We evaluated our approach on French social media data

---

[1]French language BERT models specialized on biomedical texts

related to a developmental and epileptic encephalopathy, an extremely rare disease discovered relatively recently and characterized by early-onset epilepsy (usually within the first two years of life) associated with global developmental delay (usually language and motor), intellectual disability, and frequent comorbidities such as autism spectrum disorder, and sleep disturbances(Torkamani et al., 2014; Saitsu et al., 2015). Through our experiments, we showed that our method reduces embedding anisotropy and improves the geometric structure of the embedding space, yielding better performance on an unsupervised downstream task.

Although designed and extensively validated for our specific social media dataset, we believe our Heterogeneous MultiView-Contrastive Learning (HMV-CL) framework is modular and could be applied on other rare diseases forums or social media data, sharing similar characteristics: heterogeneous textual content, sparse clinical signals, and High Dimensional-Low Sample Size (HDLSS) setting. To the best of our knowledge, no standard benchmark currently captures the unique challenges of this domain (extreme data scarcity, domain-specific jargon, mixed emotional/clinical content). While lacking formal theoretical guarantees, our empirical validation on this real-world dataset makes our approach particularly valuable as both a methodological contribution and a baseline for future work.

Overall, our work demonstrates that HMV-CL is an effective approach for representation learning in low-resource, domain-specific social media settings, and offers new perspectives for the analysis of patient-centered narratives in rare diseases.

**Paper organization** In section 2, we begin by discussing the necessary background, including on anisotropy of language embeddings, contrastive and multi-view objectives and the potential redundancy between views and subsequent dimensional collapse. In section 3, we present our proposed method before presenting our experimental setup in section 4. In section 5, we present the experimental results and discuss them.

## 2. Related Work

### 2.1. Embedding anisotropy

Recent work has highlighted that pretrained language model embeddings often exhibit strong anisotropy, where representations concentrate in narrow cones of the embedding space, reducing their discriminative capacity. This phenomenon has been extensively documented for sentence embeddings derived from BERT-like models (Ethayarajh, 2019; Li et al., 2020; Godey et al., 2024). Anisotropy negatively impacts downstream tasks such as clustering or semantic similarity estimation.

In the context of rare diseases, and particularly of families' narratives, this issue is severe because the language, mixing experiential discourse and specialized jargon, differs significantly from the general (or biomedical) corpora used during pre-training.

## 2.2. Contrastive Learning (CL)

CL is a Machine Learning paradigm that learns representations by contrasting positive pairs against negative pairs. The scientific interest in CL, measured by scientific publications and citations, has sharply increased since 2018 (Hu et al., 2024).

It has been very successful in learning representations of images and sequential data (Oord et al., 2019; Tian et al., 2020; Henaff, 2020; Chen et al., 2020; He et al., 2020) has been extended to language (Gao et al., 2021). In NLP, CL objectives have been shown to partially alleviate the issue of anisotropy. In particular, Gao et al. (2021) demonstrate that even simple contrastive setups can significantly "alleviate the anisotropy problem" in sentence embedding spaces, both empirically and theoretically.

Our framework, described in section 3 is based on Chen et al. (2020); Gao et al. (2021). In the same way, our loss pulls positive pairs together while pushing negatives apart. Compared to these frameworks, based on data augmentation to create the positive pairs, we used the complementarity of heterogeneous views as natural positive pairs.

## 2.3. Multi-view representation learning

Multi-view representation learning aims to learn by maximizing mutual information between different views of a same object. In computer vision, multi-view contrastive objectives were explored to align different visual perspectives or modalities (Tian et al., 2020). The idea has been extended to heterogeneous representations, including text, showing that contrastive alignment across views can improve robustness and generalization (Xu et al., 2023). For example, the representations can be audio and text recorded simultaneously, images and text representing the same thing, parallel text in different languages,... (Wang et al., 2015).

In biomedical and clinical domains, multi-view learning has been used to combine heterogeneous sources. For example Chen et al. (2023) used it on multi-omics data to identify cancer subtypes; Zhao et al. (2024) incorporated medical knowledge graphs and medical licensing exams to their textual data from medical records; and Bai et al. (2025) applied a MV-CL approach to MRI multi-view data and medical insights extracted from textual symptom descriptions.

## 2.4. Preventing dimensional collapse

A major risk in multi-view learning is dimensional collapse, where various projection heads learn redundant information. Collapse phenomena in SSL, i.e., when representations lose diversity, have been studied in literature. For example, He et al. (2024) show that contrastive and non-contrastive SSL methods are prone to dimensional collapse. Methods that introduce variance and decorrelation constraints to prevent such collapse in vision models (Zbontar et al., 2021; Bardes et al., 2022) have been proposed.

In the case of contrastive objectives, most of them are based on mutual information maximization (Oord et al., 2019; Tian et al., 2020; Wu et al., 2020; Zhang et al., 2020). However, maximizing mutual information alone does not prevent different views from encoding the same information. Recent work has emphasized the need to balance alignment with redundancy reduction (Wang & Isola, 2020); for example (He et al., 2024) mitigate the redundancy risk through explicit orthogonality regularization, penalizing inter-view correlations.

## 3. Method: Heterogeneous MutiView - Contrastive Learning (HMV-CL)

Our framework, illustrated in Figure 1, aims to learn a robust document-level representation by aligning heterogeneous views in a shared latent space.

### 3.1. Formal definitions and notation

We consider a batch of $N$ documents[2], each document $k \in \{1, ..., N\}$ is represented by a set of $V$ heterogeneous views.

- Input: let $x_k^{(v)} \in \mathbb{R}^{d_v}$ be the raw representation of the $v$-th view of document $k$
- Shared latent space: each view is mapped by a learnable multi-layer perceptron (MLP) followed by $L_2$-normalization to a shared space $\mathbb{R}^D$ via a view specific projection head $z_k^{(v)} = \text{Norm}\big(f_\theta^{(v)}(x_k^{(v)})\big)$ (i.e. such that $||z_k^{(v)}||_2 = 1$)
- Batch matrix representation: for a batch of $N$ documents, we define the view-specific embedding matrix $Z^{(v)} = [z_k^{(v)}]_{k \in \{1,...N\}} \in \mathbb{R}^{N \times D}$ as the stack of the projected vectors for the $v$-th view.
- $\mathbf{Z} = (Z^{(1)}, ..., Z^{(V)})$ is a tuple representing the complete state of the system for a given batch.

### 3.2. Core contrastive objective

Following Chen et al. (2020) and Gao et al. (2021)'s frameworks, we define our base contrastive objective $\ell_{CL}$ for any

---

[2]From here on "document" refers to a post from social media/online forums

two sets of projections across the batch:

$$\ell_{\text{CL}}(Z^{(i)}, Z^{(j)}) = -\frac{1}{N} \sum_{k=1}^{N} \log \frac{\exp(\frac{\text{sim}(z_k^{(i)}, z_k^{(j)})}{\tau})}{\sum_{l=1}^{N} \exp(\frac{\text{sim}(z_k^{(i)}, z_l^{(j)})}{\tau})}, \tag{1}$$

where $\text{sim}(\cdot, \cdot)$ denotes cosine similarity and $\tau > 0$ is a temperature parameter (Oord et al., 2019; Gao et al., 2021; Chen et al., 2020).

### 3.3. Alignment of multiple views

In our work, multiple complementary views capturing different aspects of the underlying narrative are associated with each document. To align representations across views, we use the average of pairwise $\ell_{CL}$ between all distinct view pairs:

$$\ell_{\text{MV-CL}}(\mathbf{Z}) = \frac{2}{V(V-1)} \sum_{\substack{i<j \\ i,j \in \{1,\ldots,V\}}} \ell_{\text{CL}}(Z^{(i)}, Z^{(j)}) \tag{2}$$

This loss encourages different views of the same document to be aligned (numerator), while maintaining separation between representations of all different documents in the batch (denominator).

### 3.4. Fusion by linear projection and global alignment

Individual view projections $z_k^{(v)} \in \mathbb{R}^D$ are concatenated per document $k$ into $[z_k^{(1)}, \ldots, z_k^{(V)}] \in \mathbb{R}^{V \times D}$.

To produce the final document-level embedding $e_k$, we apply a single learnable linear projection $g_\phi : \mathbb{R}^{V \times D} \to \mathbb{R}^D$ followed by $L_2$-normalization.

To ensure that $g_\phi$ is actively trained, we introduce an objective that forces the fused representation $e_k$ to remain consistent with its constituent views:

$$\ell_{\text{global}}(\mathbf{Z}) = \frac{1}{V} \sum_{v-1}^{V} \ell_{CL}(E, Z^{(v)}). \tag{3}$$

### 3.5. Can the heterogeneity of the views be taken as a natural regularizer?

**Multi-view collapse risk**   As seen in section 2, a contrastive objective alone on multiple views can admit degenerate solutions where projection heads learn nearly identical representations. In such cases, the views become redundant, collapsing into a single modality and failing to capture complementary information.

**Do heterogeneous views provide implicit regularization?**
Our views present substantial differences: dense contextual text embeddings (CamemBERT-v2, dim=768) vs. sparse

symbolic representations (clinical concepts/medical procedures, dim=64 and pharmacological treatments, dim=32) as described in section 4. Thus, we hypothesize that their heterogeneous nature provides a natural regularizer, making explicit orthogonality penalties unnecessary.

To test this, we implement a second variant of the loss from Equation 2: $\ell_{\text{MV-CL+OR}}$, a version of $\ell_{\text{MV-CL}}$ with orthogonality regularization $\ell_{\text{OR}}$ (He et al., 2024; Bardes et al., 2021). This loss discourages cross-view redundancy by penalizing inter-view correlation in the shared embedding space.

Let $\bar{Z}^{(i)}, \bar{Z}^{(j)} \in \mathbb{R}^{N \times D}$ denote the matrices of normalized embeddings. The orthogonality loss is defined as (He et al., 2024; Bardes et al., 2021):

$$\ell_{\text{OR}}(\mathbf{Z}) = \frac{2}{V(V-1)} \sum_{\substack{i<j \\ i,j \in \{1,\ldots V\}}} \left\| \frac{1}{N}(\bar{Z}^{(i)})^\top \bar{Z}^{(j)} \right\|_F^2, \tag{4}$$

where $\|\cdot\|_F$ denotes the Frobenius norm.

The full $\ell_{\text{MV-CL+OR}}$ training objective combines multi-view contrastive alignment with orthogonality regularization:

$$\ell_{\text{MV-CL+OR}} = \ell_{\text{MV-CL}} + \lambda \, \ell_{\text{OR}} \tag{5}$$

where $\lambda > 0$ controls the trade-off between alignment and diversity across views.

## 4. Experimental setup

### 4.1. Data

**Context and cohort description**   We conducted our experiments on a corpus of 784 (1,657 before preprocessing) unlabeled French-language social media posts related to a developmental and epileptic encephalopathy, a very rare disease where young children are affected by epilepsy crises (cf. section 1). The dataset consists of authentic narratives from families, collected from a patients' association's [3] social media platforms (Facebook, Facebook Messenger, Whatsapp) between August 2021 and December 2025. The mean number of posts by user is 10.45 (median=4; min=1;max=144); their mean length (whitespace separation tokens) is 73.45 (median=41.5; min=3; max=1509); and their mean length (CamemBERT-v2 subword tokens) is 97.76 (median=56; min=7; max=1024). The huge variability of number of posts per user is mainly due to the fact that some users have been in the groups for 4.5 years while others only a few days/weeks.

**Data and code protection and availability and regulatory compliance**   The collection and treatment of the data is in compliance with the French legal framework of MR-004

---

[3]kept anonymous for the blind review process

set up by the CNIL[4]. Families in the different groups have been informed of the re-use of their data and of their rights regarding this data; they have the possibility to opt-out of the study at any time. Their privacy is also protected by the pseudonymization of all messages by replacing every name or surname by tags, including the names of doctors or any other person mentioned. Because of the high sensitivity (cf. section 6) of this data and the risk of reidentification, the dataset cannot be shared publicly. For research purposes, and with a solid security framework, access to the data can be discussed with the patients' organization.

However, to ensure the reproducibility of our results, the code (including the multi-view encoders and evaluation metrics) code of our HMV-CL framework is provided in the supplementary material and will be released on GitHub upon publication.

**Preprocessing**  As stated previously, each document is pseudonymized by replacing names by a tag. All the links are also tagged, to reduce noise. Additionnally, a fine-tuned BERT classifier (and human annotation) allowed to select only the posts containing a narrative of a lived experience of the disease, the 784 used in this work.

**Construction of the views**  Each document corresponds to an unstructured narrative and has been associated with up to four different views and minimally one: (i) the raw free-text representation obtained with `CamemBERT-v2` (mean pooling) available for 100% of the documents (Antoun et al., 2024), as well as embeddings derived from domain-specific lexicons (at least one for 38.5% of the documents) from SNOMED-CT (El-Sappagh et al., 2018), (ii) clinical concepts, (iii) medical procedures and (iv) pharmacological treatments.

*Table 1.* Views and their dimensions

| VIEW | FEATURE DIM. |
|---|---|
| TEXT | 768 |
| CLINICAL CONCEPTS | 64 |
| MEDICAL PROCEDURES | 32 |
| PHARMACOLOGICAL TREATMENTS | 64 |

For each symbolic view, we set the embedding dimension to approximately $\sqrt{|L|}$, where $|L|$ is the lexicon size (Table 1). This heuristic provides a compact yet expressive representation that compresses the discrete one-hot space into a low-dimensional and continuous space while limiting over-parameterization in our HDLSS setting. Across all views, the total embedding dimensionality is approx-

---

[4]*Commission nationale de l'informatique et des libertés*, French national data protection authority

imately equal to the number of messages (dimension-to-sample ratio $\approx 1.14$), which is relatively modest for high-dimensional text representations and helps mitigate over-fitting compared to more extreme HDLSS settings. This dimensionality is large enough to capture meaningful similarity structure between views, yet small enough to limit over-parameterization in our HDLSS setting.

## 4.2. Evaluation

### 4.2.1. EVALUATION OF THE EMBEDDING SPACE

This evaluation focuses on some of the properties of the embeddings spaces, independently of any downstream task, although they do have an impact on the usability for downstream tasks such as clustering, retrieval and semantic exploration. All the following metrics are averaged over 15 random training runs.

- Uniformity: this metric evaluates how evenly embeddings are distributed accross the $L_2$-normalized hypersphere (Wang & Isola, 2020). A higher value indicates a "collapsed" distribution where vectors occupy a limited area.
- Isotropic Coefficient: we measured embedding isotropy using the ratio between the smallest and largest singular values of the centered embedding matrix (Wang et al., 2020; Mu et al., 2017). It ranges from 0 to 1 and the closer to zero, the more anisotropic the embedding space. However, it is limited by its high sensitivity to non-informative dimensions.
- Cosine similarity distribution: we reported the mean of the cosine similarities of the embeddings of 5,000 randomly sampled pairs of documents. In anisotropic spaces, the cosine similarities are high, leading to poor discrimination between documents.
- Effective Rank: this metric uses the Shannon entropy of the singular value spectrum to determine the number of actually useful dimensions (Garrido et al., 2023), that we reported alongside the number of dimensions of the embedding considered.
- Explained variance: we computed the proportion of variance explained by the first ten principal components using Principal Component Analysis (PCA) (Pearson, 1901). Indeed, in anisotropic spaces the variance is concentrated along a few dominant directions, while in isotropic spaces variance is not concentrated on a few directions, but better balanced.

### 4.2.2. EVALUATION ON A DOWNSTREAM UNSUPERVISED TASK

Given document embeddings, we used `BERTopic`'s framework (Grootendorst, 2022) to cluster the embeddings, and computed averages and standard deviations of usual clustering metrics over 15 runs.

- Silhouette score: it measures how well each document lies within its assigned cluster compared to other clusters (Rousseeuw, 1987). Higher values indicate better cluster separation. For a point $i$, let $a(i)$ be the mean intra-cluster distance and $b(i)$ be the mean nearest-cluster distance, then: $s_i = \frac{b(i)-a(i)}{\max(a(i),b(i))}$ and the score given is the mean over all samples $i$.
- Davies-Bouldin index: it evaluates the average similarity between each cluster and its most similar one (Davies & Bouldin, 1979). Lower values indicate more compact and well-separated clusters. Let $S_k$ be the intra-cluster disperson of cluster $k$ and $M_{ij}$ be the distance between centroids of clusters $i$ and $j$, then: $DB = \frac{S_i+S_j}{M_{ij}}$
- Adjusted Rand Index: it measures the similarity between two data clusterings, corrected for chance (Hubert & Arabie, 1985). ARI$= 1$ indicates perfect match, while ARI$= 0$ indicates random labeling.
- Outliers Ratio: it measures the percentage of documents that the clustering algorithm fails to assign to any specific thematic cluster, effectively treating them as noise. In our framework, this metric serves as a key indicator of clustering stability.

### 4.3. Training details

We trained the multi-view contrastive model using the AdamW optimizer (Loshchilov & Hutter, 2019) with a learning rate of $1e^{-4}$ and weight decay of $1e^{-5}$. The training is performed for a fixed number of epochs of 50 and minibatches of size 32 are used. Each experiment is repeated over 15 random seeds to ensure robustness and all reported results are averaged over 15 runs.

### 4.4. Ablation studies

- Effect of multi-view supervision: throughout the study, we compared HMV-CL with or without orthogonality regularization (OR) against a text-only baseline using `CamemBERT-v2` embeddings (Antoun et al., 2024). This isolates the impact of incorporating the three other views (clinical concepts, medical procedures and pharmacological treatments).

- Effect of temperature ($\tau$): we evaluated HMV-CL across different temperature values ($\tau \in \{0.025, 0.05, 0.1, 0.2, 0.5, 1\}$) on both embedding geometry and downstream clustering.

- Effect of orthogonality regularization ($\lambda$): we evaluated a variant of the model trained with an orthogonality regularizer (HMV-CL+OR) as described in subsection 3.5). We reported the results of this variant, and also varied the weight ($\lambda \in \{0, 0.1, 0.5, 1, 5, 10\}$) for $\tau = 0.1$ and $\tau = 1$.

## 5. Results and discussion

### 5.1. The geometry-performance trade-off

Table 2 presents the metrics of both embedding space geometry and downstream clustering. All settings of HMV-CL proposed here -with or without orthogonality regularization (OR) and with $\tau = 0.1$ or $\tau = 1$- improve the baseline, both on geometric and downstream metrics. However, a trade-off between geometry and downstream performance appears in the results.

**Improved geometry of the embedding spaces, especially for low temperature** Anisotropy is reduced with our different settings of HMV-CL, as illustrated by the sharp decrease of uniformity (around $-3$ or $-2$ in all settings compared to $+\infty$ for the baseline), the increase of Isotropic Coefficient (IC) (0.053 compared to $8.25 \times 10^{-9}$ for $\tau = 0.1$) and the decrease of average cosine similarities (0.03 compared to 0.88 for $\tau = 1$).

Moreover, while the text-only baseline utilizes only a fraction of its available space (effective rank of 209 for a 768 dimensions space, i.e. $\approx 27\%$), our framework demonstrates significantly higher efficiency , utilizing 45 out of 64 dimensions ($\approx 70\%$) for $\tau = 0.1$ and about $46\%$ for $\tau = 1$ with orthogonality regularization (OR) (cf. Table 2).

The distribution of variance across Principal Components (PCs) further clarifies how information is organized witin the shared latent space. For $\tau = 0.1$, the variance is broadly distributed, with the top-1 PC acounting for only $7.73\%$ of variance and top-10 PCs for just more than half of it ($59.5\%$). On the other hand, for $\tau = 1$, the top-10 PCs concentrate almost all of the variance ($96.18\%$) and the top-1 jumps to $35.13\%$ (cf. Table 2).

With $\tau = 1$, although the geometry metrics are degraded compared to $\tau = 0.1$, they remain improved compared to baseline.

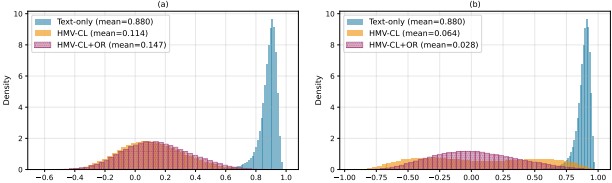

*Figure 2.* Cosine distributions between 5,000 randomly sampled pairs of embeddings for baseline, HMV-CL and HMV-CL+OR for (a) $\tau = 0.1$ and (b) $\tau = 1$. ($\lambda = 10$)

Figure 2 shows cosine similarities distributions of $5,000$ randomly sampled pairs of embeddings for each setting, with $\tau = 0.1$ (Figure 2 (a)) and with $\tau = 1$ (Figure 2 (b)). The text-only baseline exhibits a sharp and narrow peak

*Table 2.* Embedding space & clustering metrics (mean $\pm$ std, 15 runs). $\lambda = 10$. The best and second best results are highlighted in bold and underlined respectively. For text-only, no std reported as the embeddings are deterministic.

|  | TEXT-ONLY | HMV-CL $\tau = 0.1$ | HMV-CL+OR $\tau = 0.1$ | HMV-CL $\tau = 1$ | HMV-CL+OR $\tau = 1$ |
|---|---|---|---|---|---|
| **EMBEDDING SPACE GEOMETRY** |  |  |  |  |  |
| UNIFORMITY ↓ | $\infty$ | $\mathbf{-3.17 \pm 0.04}$ | $\underline{-3.03 \pm 0.05}$ | $-2.39 \pm 0.08$ | $-3.02 \pm 0.04$ |
| ISOTROPIC COEFFICIENT ↑ | $8.25 \cdot 10^{-9}$ | $\mathbf{0.053 \pm 0.004}$ | $\underline{0.052 \pm 0.002}$ | $0.009 \pm 0.001$ | $0.017 \pm 0.001$ |
| AVG.COS.SIM. ↓ | 0.88 | $0.10 \pm 0.01$ | $0.14 \pm 0.01$ | $\underline{0.05 \pm 0.01}$ | $\mathbf{0.03 \pm 0.01}$ |
| EFFECTIVE RANK (DIM) | 209 (768) | $45 \pm 0.3$ (64) | $45 \pm 0.3$ (64) | $26 \pm 0.6$ (64) | $30 \pm 0.6$ (64) |
| TOP-10 PCS VAR. (%) | 51.67 | $59.50 \pm 1.06$ | $58.16 \pm 0.96$ | $96.18 \pm 0.33$ | $92.64 \pm 0.98$ |
| TOP-1 PCS VAR. (%) | 16.20 | $7.73 \pm 0.22$ | $8.19 \pm 0.17$ | $35.13 \pm 1.59$ | $16.45 \pm 0.67$ |
| **DOWNSTREAM UNSUPERVISED TASK** |  |  |  |  |  |
| SILHOUETTE ↑ | $\underline{0.17}$ | $0.08 \pm 0.01$ | $0.10 \pm 0.001$ | $\mathbf{0.25 \pm 0.02}$ | $0.12 \pm 0.09$ |
| DAVIES-BOULDIN ↓ | 1.79 | $1.65 \pm 0.10$ | $1.58 \pm 0.06$ | $\mathbf{1.04 \pm 0.07}$ | $\underline{1.28 \pm 0.09}$ |
| ADJUSTED RAND INDEX ↑ | - | $0.71 \pm 0.38$ | $\underline{0.72 \pm 0.39}$ | $\mathbf{0.81 \pm 0.07}$ | $0.18 \pm 0.16$ |
| OUTLIERS RATIO (%) ↓ | 95.15 | $12.07 \pm 4.78$ | $\underline{11.49 \pm 4.49}$ | $\mathbf{10.59 \pm 3.90}$ | $34.11 \pm 17.85$ |

centered at a mean similarity of 0.88. This provides visual evidence of the anisotropy issue of this embedding space. For $\tau = 0.1$, we remark a broad distribution spreading significantly compared to the baseline around the mean of approximately 0.1. We also note that OR seems to have no effect, as the curves are almost indiscernible, much as the results in Table 2. For $\tau = 1$, the mean similarity shifts to 0.06 (without OR) or 0.03 (with OR). Here, just as the results in Table 2 suggest, the OR has more effect than for $\tau = 0.1$.

**Better and more stable clustering, especially for high temperature** The geometric improvements translate into significant gains in clustering performance. Especially with $\tau = 1$, all metrics reported in Table 2 are drastically improved compared to baseline, which despite a silhouette score of 0.17 suffers from an outliers ratio of 95.15%. In comparison, HMV-CL with $\tau = 1$ reduces outliers ratio to just 10.59%, demonstrating that the framework successfully extracts information from the reduced latent space, where families' narratives can be grouped more easily.

**Visual analysis of the embeddings spaces** Uniform Manifold Approximation and Projection (UMAP) (Healy & McInnes, 2024) visualizations (Figure 3) confirm improvements observed through the metrics of Table 2: text-only embeddings (left) collapse into a single cloud with points concentrated in a limited range on both UMAP dimensions ($\approx [2, 12] \times [4, 12]$). In contrast, our embeddings (center and right) occupy much larger regions, $\approx [-5, 15] \times [-5, 12]$ for $\tau = 1$ (Figure 3 (b), center). We observe what will be discussed in subsection 5.2: for $\tau = 0.1$, OR has little impact, while it does have an important impact for $\tau = 1$.

**5.2. Natural regularization vs explicit constraints**

**View heterogeneity is a sufficient regularizer for $\tau = 0.1$** Figure 4 displays the cosine similarity matrices between view representations for HMV-CL and HMV-CL+OR. We observe that the correlation between the three added views are very low (between 0.00 and 0.32 without OR) and that the correlation between text and these views is sensibly higher (between 0.09 and 0.48 without OR). Figure 4 (b) confirms that OR decorrelates the views, for example, text-pharma (from 0.48 to 0.17) and text-clinical (from 0.32 to 0.11). The low correlations between views provide sufficient diversity to prevent dimensional collapse without OR. Indeed, for $\tau = 0.1$, HMV-CL maintains similar variance structures to HMV-CL+OR, which reinforces the hypothesis that the heterogeneous nature of clinical views acts as a natural regularizer, preventing total collapse while allowing the model to focus information along a set of more informative, clusterable directions.

**OR paradox for $\tau = 1$** The ablation study of the OR weight $\lambda$ in Appendix B reveals that the impact of explicit regularization is temperature-dependent. For $\tau = 0.1$, the space is already highly uniform, and varying $\lambda$ has almost no effect on the effective rank or isotropic coefficient (IC), which both stay stable. It also has limited effect on uniformity (-3.17 to -3.03).

However, for $\tau = 1$, OR has a much more pronounced effect on geometry: increasing $\lambda$ from 0 to 10 successfully pushes the effective rank from 26 to 30 and improves the IC from 0.010 to 0.017, and reduces uniformity from -2.40 to -3.02. Despite these improved geometric metrics, explicit OR seems undesirable for task performance. Forcing higher orthogonality at this temperature causes the ARI to collapse from 0.81 down to 0.18, silhouette to decrease from 0.26 to 0.12 and nearly triples the outliers ratio (from 10.59% to 34.11%).

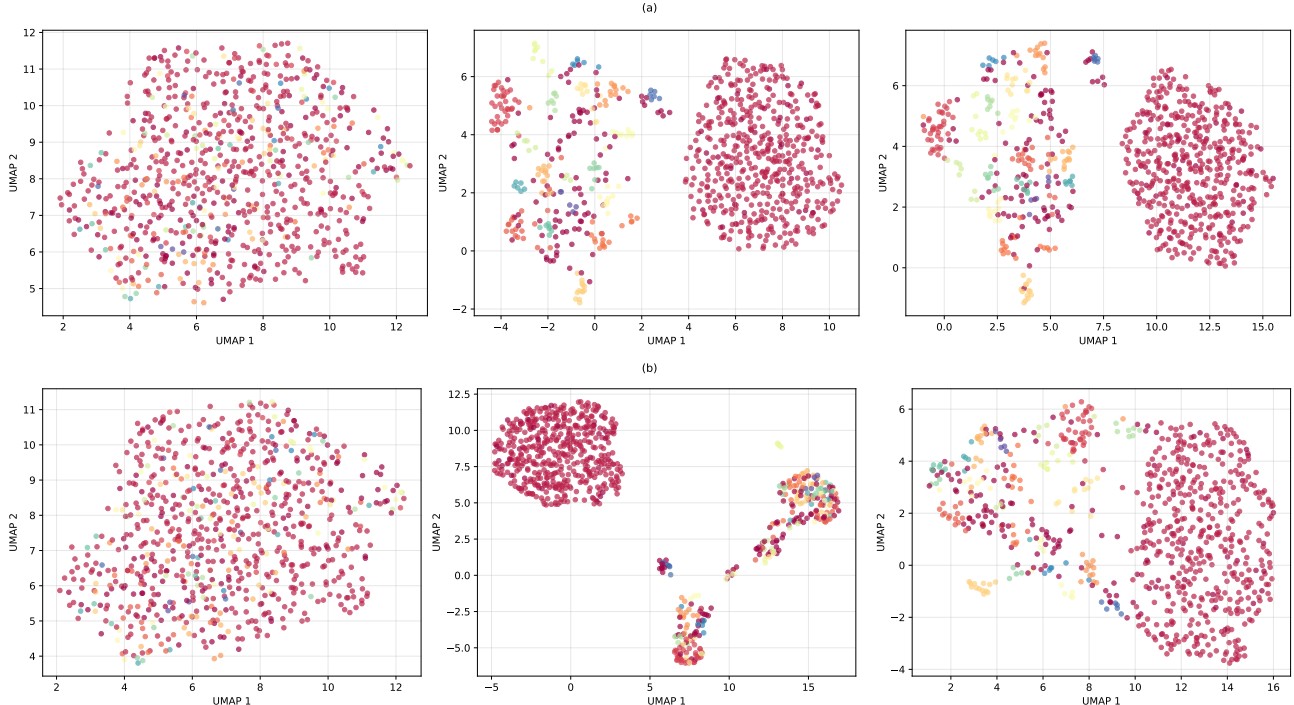

*Figure 3.* UMAP projections of document embeddings. Points are colored by BERTopic cluster assignement. From left to right: `CamemBERT-v2` embeddings, HMV-CL embeddings, HMV-CL+OR ($\lambda = 10$) embeddings. (a) $\tau = 0.1$. (b) $\tau = 1$. *Note: these visualizations are provided for qualitative assessment of cluster density and distribution; all quantitative metrics are computed directly in the shared latent space $\mathbb{R}^{64}$ to avoid the distortion inherent in non-linear dimensionality reduction*

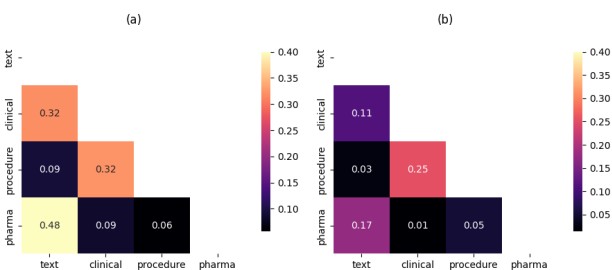

*Figure 4.* Cosine similarity matrices between view representations for (a) HMV-CL and (b) HMV-CL+OR ($\tau = 1, \lambda = 10$)

This confirms that while OR can remedy to the geometric losses of higher temperature (it improves the effective rank and IC), it comes at the expense of the semantic clusters that are crucial for analyzing patient trajectories. The natural regularization provided by the heterogeneous views is therefore sufficient and preferred for real-world applications.

## 6. Conclusion

Overall, HMV-CL considerably improves embedding space geometry in our specific HDLSS setting of rare disease families' posts, enabling *in fine* the analysis of these posts in

a public health objective. Indeed, we were able to reduce anisotropy of the embedding space and the heterogeneity of views prevented collapse naturally, avoiding the views to learn redundant information as can be observed in other contrastive settings. Finally, we drastically improved downstream clusterability of the embeddings.

However, there are some important limitations to this study. Firstly, our evaluation focuses exclusively on a single rare disease cohort (a developmental and epileptic encephalopathy) in French language, limiting immediate generalizability to other patient populations. Moreover, the absence of standardized benchmarks presenting similar specificities does not allow direct comparison with other methods. Finally, while empirically robust, HMV-CL lacks formal convergence guarantees.

Future work could take multiple directions. We could extend HMV-CL to cross-lingual rare disease forums, investigate adaptive view weighting to further optimize the inherent regularization provided by view heterogeneity. We could also look into formal convergence guarantees. We will also apply our framework to concretely analyze patient trajectories and needs.

## Impact Statement

HMV-CL enables more effective analysis of rare disease patients' and families' narratives from social media, potentially accelerating the understanding of their needs, as well as natural histories for underserved medical conditions. By improving representation learning in low-resource HDLSS settings, our framework facilitates patient-centered NLP applications.

However, we are fully aware of the severe risks of data misuse such as malicious repurpose of these families' and patients' narratives for scams, blackmail or targeted misinformation campaigns. There is also a risk of this data or the results of downstream studies informing discriminatory healthcare policies or resource allocation decisions, affecting vulnerable populations. These risks are why deployment of such methods and use of such data require strict patient privacy safeguards (including but not limited to GDPR compliance) and minutious monitoring by the primary stakeholders (here rare disease families).

The method processes pseudonymized French social media data, and was developed hand in hand with a patients' association, especially to monitor downstream conclusions and ensure the families included are well informed and can easily assert their rights regarding their data.

Another risk is the potential misinterpretation of unverified social media claims as medical evidence; which is why this work is also supervised by clinicians.

Overall, HMV-CL democratizes rare disease research by establishing reproducible baselines for patient-driven discovery without requiring massive labeled datasets.

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

## A. Temperature ablation study

Table 3. $\tau$ ablation study (mean $\pm$ std, 15 runs). $\lambda = 0$. The best and second best results are highlighted in bold and underlined respectively.

| $\tau$ | 0.025 | 0.05 | 0.1 | 0.2 | 0.5 | 1 |
|---|---|---|---|---|---|---|
| **EMBEDDING SPACE GEOMETRY** | | | | | | |
| UNIFORMITY ↓ | $-2.60 \pm 0.04$ | $-2.98 \pm 0.02$ | $\underline{-3.17 \pm 0.04}$ | $\mathbf{-3.26 \pm 0.06}$ | $-2.98 \pm 0.07$ | $-2.40 \pm 0.10$ |
| ISOTROPIC COEFFICIENT ↑ | $0.039 \pm 0.003$ | $\underline{0.052 \pm 0.004}$ | $\mathbf{0.053 \pm 0.004}$ | $0.036 \pm 0.002$ | $0.018 \pm 0.001$ | $0.010 \pm 0.001$ |
| AVG.COS.SIM. ↓ | $0.23 \pm 0.01$ | $0.15 \pm 0.01$ | $0.10 \pm 0.01$ | $0.06 \pm 0.01$ | $\mathbf{0.04 \pm 0.01}$ | $\underline{0.05 \pm 0.01}$ |
| EFFECTIVE RANK | $42 \pm 0.4$ | $45 \pm 0.3$ | $45 \pm 0.4$ | $41 \pm 0.4$ | $33 \pm 0.5$ | $26 \pm 0.7$ |
| TOP-10 PCS VAR. (%) | $73.74 \pm 0.85$ | $64.78 \pm 0.96$ | $60.12 \pm 1.08$ | $65.73 \pm 1.19$ | $87.57 \pm 1.32$ | $96.11 \pm 0.33$ |
| TOP-1 PCS VAR. (%) | $13.12 \pm 0.59$ | $8.83 \pm 0.31$ | $7.67 \pm 0.24$ | $9.71 \pm 0.38$ | $19.42 \pm 0.95$ | $35.20 \pm 1.74$ |
| **DOWNSTREAM UNSUPERVISED TASK** | | | | | | |
| SILHOUETTE ↑ | $0.10 \pm 0.01$ | $0.09 \pm 0.01$ | $0.07 \pm 0.01$ | $0.07 \pm 0.01$ | $\underline{0.11 \pm 0.02}$ | $\mathbf{0.26 \pm 0.02}$ |
| DAVIES-BOULDIN ↓ | $1.53 \pm 0.08$ | $1.58 \pm 0.08$ | $1.63 \pm 0.07$ | $1.57 \pm 0.04$ | $\underline{1.26 \pm 0.07}$ | $\mathbf{1.02 \pm 0.07}$ |
| ADJUSTED RAND INDEX ↑ | $\mathbf{0.86 \pm 0.03}$ | $0.82 \pm 0.31$ | $0.77 \pm 0.30$ | $0.79 \pm 0.08$ | $0.79 \pm 0.11$ | $0.74 \pm 0.12$ |
| OUTLIERS RATIO (%) ↓ | $18.83 \pm 2.28$ | $\underline{13.44 \pm 3.75}$ | $13.75 \pm 4.17$ | $16.02 \pm 2.73$ | $15.74 \pm 4.35$ | $\mathbf{13.37 \pm 6.46}$ |

## B. Orthogonality Regularization weight ablation study

Table 4. $\lambda$ ablation study (mean $\pm$ std, 15 runs). $\tau = 1$. The best and second best results are highlighted in bold and underlined respectively.

| $\lambda$ | 0 | 0.1 | 0.5 | 1 | 5 | 10 |
|---|---|---|---|---|---|---|
| **EMBEDDING SPACE GEOMETRY** | | | | | | |
| UNIFORMITY ↓ | $-2.40 \pm 0.10$ | $-2.43 \pm 0.10$ | $-2.54 \pm 0.10$ | $-2.64 \pm 0.09$ | $\underline{-2.92 \pm 0.05}$ | $\mathbf{-3.02 \pm 0.04}$ |
| ISOTROPIC COEFFICIENT ↑ | $0.010 \pm 0.001$ | $0.010 \pm 0.001$ | $0.011 \pm 0.001$ | $0.012 \pm 0.001$ | $\underline{0.015 \pm 0.001}$ | $\mathbf{0.017 \pm 0.001}$ |
| AVG.COS.SIM. ↓ | $0.05 \pm 0.01$ | $0.05 \pm 0.01$ | $0.05 \pm 0.01$ | $0.05 \pm 0.01$ | $\underline{0.04 \pm 0.01}$ | $\mathbf{0.03 \pm 0.01}$ |
| EFFECTIVE RANK | $26 \pm 0.7$ | $26 \pm 0.7$ | $26 \pm 0.7$ | $28 \pm 0.6$ | $30 \pm 0.6$ | $30 \pm 0.6$ |
| TOP-10 PCS VAR. (%) | $96.11 \pm 0.33$ | $96.01 \pm 0.34$ | $95.61 \pm 0.39$ | $95.14 \pm 0.47$ | $93.13 \pm 0.93$ | $92.64 \pm 0.98$ |
| TOP-1 PCS VAR. (%) | $35.20 \pm 1.74$ | $34.27 \pm 1.70$ | $31.02 \pm 1.59$ | $28.06 \pm 1.46$ | $19.22 \pm 0.81$ | $16.45 \pm 0.67$ |
| **DOWNSTREAM UNSUPERVISED TASK** | | | | | | |
| SILHOUETTE ↑ | $\mathbf{0.26 \pm 0.02}$ | $\underline{0.25 \pm 0.02}$ | $0.21 \pm 0.04$ | $0.16 \pm 0.04$ | $0.12 \pm 0.05$ | $0.12 \pm 0.09$ |
| DAVIES-BOULDIN ↓ | $\mathbf{1.02 \pm 0.07}$ | $\underline{1.04 \pm 0.07}$ | $1.11 \pm 0.07$ | $1.17 \pm 0.04$ | $1.22 \pm 0.05$ | $1.28 \pm 0.09$ |
| ADJUSTED RAND INDEX ↑ | $\mathbf{0.81 \pm 0.07}$ | $\underline{0.73 \pm 0.13}$ | $0.74 \pm 0.09$ | $0.59 \pm 0.24$ | $0.43 \pm 0.17$ | $0.18 \pm 0.16$ |
| OUTLIERS RATIO (%) ↓ | $\mathbf{10.59 \pm 3.90}$ | $\underline{14.30 \pm 7.12}$ | $13.93 \pm 4.31$ | $15.64 \pm 6.43$ | $29.35 \pm 12.03$ | $34.11 \pm 17.85$ |

Table 5. $\lambda$ ablation study (mean $\pm$ std, 15 runs). $\tau = 0.1$. The best and second best results are highlighted in bold and underlined respectively.

| $\lambda$ | 0 | 0.1 | 0.5 | 1 | 5 | 10 |
|---|---|---|---|---|---|---|
| **EMBEDDING SPACE GEOMETRY** | | | | | | |
| UNIFORMITY ↓ | $-3.17 \pm 0.04$ | $-3.17 \pm 0.04$ | $-3.16 \pm 0.04$ | $-3.15 \pm 0.04$ | $\underline{-3.08 \pm 0.05}$ | $\mathbf{-3.03 \pm 0.05}$ |
| ISOTROPIC COEFFICIENT ↑ | $0.052 \pm 0.004$ | $0.053 \pm 0.004$ | $0.053 \pm 0.003$ | $0.053 \pm 0.003$ | $0.052 \pm 0.003$ | $0.052 \pm 0.002$ |
| AVG.COS.SIM. ↓ | $0.10 \pm 0.01$ | $\underline{0.10 \pm 0.01}$ | $\mathbf{0.10 \pm 0.01}$ | $0.11 \pm 0.01$ | $0.12 \pm 0.01$ | $0.14 \pm 0.01$ |
| EFFECTIVE RANK | $45 \pm 0.4$ | $45 \pm 0.4$ | $45 \pm 0.3$ | $45 \pm 0.3$ | $45 \pm 0.3$ | $45 \pm 0.3$ |
| TOP-10 PCS VAR. (%) | $60.12 \pm 1.08$ | $60.05 \pm 1.06$ | $59.38 \pm 1.03$ | $59.26 \pm 1.05$ | $58.60 \pm 1.03$ | $58.16 \pm 0.96$ |
| TOP-1 PCS VAR. (%) | $7.67 \pm 0.24$ | $7.66 \pm 0.25$ | $7.70 \pm 0.21$ | $7.69 \pm 0.20$ | $7.84 \pm 0.18$ | $8.19 \pm 0.17$ |
| **DOWNSTREAM UNSUPERVISED TASK** | | | | | | |
| SILHOUETTE ↑ | $0.07 \pm 0.01$ | $0.07 \pm 0.01$ | $0.08 \pm 0.01$ | $0.08 \pm 0.01$ | $\underline{0.09 \pm 0.01}$ | $\mathbf{0.10 \pm 0.01}$ |
| DAVIES-BOULDIN ↓ | $1.63 \pm 0.07$ | $1.65 \pm 0.06$ | $1.62 \pm 0.06$ | $1.63 \pm 0.06$ | $\underline{1.61 \pm 0.07}$ | $\mathbf{1.58 \pm 0.06}$ |
| ADJUSTED RAND INDEX ↑ | $0.71 \pm 0.38$ | $0.78 \pm 0.30$ | $\mathbf{0.82 \pm 0.31}$ | $\underline{0.82 \pm 0.31}$ | $0.72 \pm 0.39$ | $0.72 \pm 0.39$ |
| OUTLIERS RATIO (%) ↓ | $13.75 \pm 4.17$ | $13.28 \pm 4.21$ | $12.76 \pm 3.52$ | $12.80 \pm 3.48$ | $\underline{11.85 \pm 4.57}$ | $\mathbf{11.49 \pm 4.49}$ |

