# OpenReview forum: "HMV-CL: Heterogeneous Multi-View Contrastive Learning for Improved Representation of Rare Disease Patient Narratives"
_ICML.cc/2026/Conference — Submitted to ICML 2026_

### Official Review · Reviewer_SC8p · 2026-03-11

**Soundness:** 3
**Presentation:** 3
**Significance:** 3
**Originality:** 3
**Overall Recommendation:** 4
**Confidence:** 4

**Summary:**

The authors propose a method to improve the extraction of textual data together with other modalities, focusing on low sample data regimes, such as in rare diseases. The core of the method uses a multi-view contrastive learning approach that aligns the dense embeddings from CamemBERT-v2 with the representations from specific views, such as clinical signs and medical procedures. Specifically, the authors use a contrastive loss objective across the views and patients, as well as another global loss for the final fusion step. The authors then evaluate on a real-world dataset from developmental and epileptic encephalopathy, showing across multiple metrics that the embeddings improve.

**Compliance With Llm Reviewing Policy:**

Affirmed.

**Final Justification:**

I update my soundness score to 3 and my overall recommendation to a Weak Accept for this paper. The authors put forth a commendable, good-faith effort during the rebuttal phase, successfully clarifying the secondary dataset statistics and providing a comprehensive batch-size ablation study that transparently corrected a previous error in their claims. This academic honesty and thoroughness significantly strengthened my confidence in the submission's core findings. However, the rebuttal fell slightly short of being comprehensive, it missed providing the complete suite of geometric metrics for the domain-specific baselines, and it bypassed the requested deeper linguistic analysis. While the downstream metrics provided for the general health dataset were helpful, the underlying hypothesis regarding vocabulary differences requires more results. In future work, the authors should conduct a more rigorous general health dataset vocabulary analysis to truly substantiate their claims. Examples of this could include calculating out-of-vocabulary (OOV) rates between the rare disease and general datasets, measuring the actual lexical overlap with standard pretraining corpora, or analyzing specific domain-term frequencies. Despite these gaps in empirical comparison, the core methodological contributions for low-resource, multi-view clinical data are sound, and I believe the strengths of the paper warrant acceptance.

**Key Questions For Authors:**

1. I appreciate that the single-dataset focus was noted in the limitations. However, to prove the generalizability of the HMV-CL framework, could the authors provide results on a secondary dataset (even if synthetic or less rare)?
2. Given that CamemBERT-bio and DrBERT were highlighted in the introduction as relevant domain-specific models, why were they not included as baselines alongside CamemBERT-v2?
3. Can you summarize which hyperparameter settings should be used initially?
4. Since the contrastive loss relies heavily on in-batch negatives, how sensitive is the model to the batch size of 32?

If the authors can provide results on other datasets, more text embeddings models and the further information on hyperaparameters as well as the effect of batch sampling, I would be willing to raise my Soundness score and overall recommendation.

**Limitations:**

Yes.

**Strengths And Weaknesses:**

Soundness:
- Strength: The method combines several well-known techniques into a novel overall approach that appears to be technically sound.
- Strength: The authors extensively analyze the resulting embeddings on the developmental and epileptic encephalopathy dataset.
- Weakness: While the authors explicitly acknowledge in their limitations that evaluating only one cohort limits generalizability , the lack of evaluation on even a synthetic dataset remains a significant weakness in proving the method's robustness.
- Weakness: The paper evaluates only the general-purpose CamemBERT-v2 model. Despite explicitly mentioning domain-specific models like CamemBERT-bio and DrBERT in the introduction , these are excluded from the experimental setup, making it difficult to assess architectural or domain-specific fluctuations.
- Weakness: The authors mention using a mini-batch size of 32 , but they do not discuss how the batch setup impacts the in-batch negative pairs, which is a known vulnerability in contrastive learning objectives.

Presentation:
- Strength: The introduction, related work and methods section are generally clearly written and well structured.
- Weakness: The different variations of the model are hard to follow in the results sections, especially with different hyperparameters and settings. This results in it being hard to derive the conclusions, especially as to which setting is optimal.
- Weakness: Some minor aspects are missing, such as the end of the figure 1 description, and the definition of E in equation 3.

Significance:
- Strength: Addressing low-data regimes with high dimensionality is a critical issue in biomedical data, especially for rare or pediatric diseases.
- Weakness: Given that the method was benchmarked only on one dataset and with no other baselines, except the text only approach, it is hard to understand the significance.

Originality:
- Strength: The proposed method appears to be novel, and provides a new yet important task.
- Weakness: While the overall approach to this specific problem is novel, the individual methodological components are largely based on existing techniques.

---

> ### Author Rebuttal · Authors · 2026-03-30
>
> We are grateful for your constructive feedback, and detailed strengths and weaknesses of every aspect of the paper. We will make sure to fix the missing end of figure 1 legend and definition of E in equation 3.
>
> **Secondary datasets and tasks.** We have conducted a second experiment the public dataset from Naseem et al. (2022)'s paper*. This dataset consists of 10,015 Reddit posts written in English and coming from subreddits where users are likely to discuss topics related to health in general (not rare conditions). Our method ($\tau=1$/no OR) improves geometry metrics compared to the baseline (roberta-base): mean cosine similarity (0.01 compared to 0.95), IC ($1.4 \times 10^{-3}$ compared to $8.6 \times 10^{-8}$) and uniformity (-3.37 compared to $\infty$).
>
> The downstream task associated to this dataset is the classification of these posts into three balanced categories (personal health mentions, non-personal and figurative/hyperbolic). We used a simple MLP trained using AdamW optimizer to evaluate the discriminative capacity of our embeddings (train/test split of 0.8/0.2). We achieved similar performance with the baseline embeddings and ours, with macro F1 of 0.69 and 0.70 respectively.
>
> We did the same experiment (same simple architecture, same train/test split, same optimizer) on a classification task on another dataset consisting of 1,095 labeled (by two annotators) messages mentioning various developmental and epileptic encephalopathies on social media. The objective was to differentiate mentions of personal experience of the disease and other discourses (imbalanced dataset). In this HDLSS context, our embeddings ($\tau=1$/no OR) significantly outperformed the baseline on the downstream task (supervised classification), increasing the weighted-average F1 from 0.77 to 0.86.
>
> These additional results indicate that while standard pretrained models perform adequately on "general health" social media posts (likely because the vocabulary is more common and the data distribution is closer to their pretraining corpora), the added value of HMV-CL is most pronounced in highly specialized domains. They also show that our framework generalizes well to less rare context and other languages than French.
>
> **Domain-specific baselines.** As pointed out by the reviewer, CamemBERT-bio and DrBERT are mentioned in the introduction and could have been used as alternative baselines. We added these experiments to our initial CamemBERT-v2 baseline. In both cases, uniformity is $\infty$ like for CamemBERT-v2 and IC is in the same order of magnitude ($10^{-9}$). Regarding effective rank, CamemBERT-bio utilizes 23\% of the available space (close to the 27\% of CamemBERT-v2) while DrBERT utilizes close to 40\%. Downstream task performance metrics are in the same range as our initial baseline. All these figures can be included as an extra appendix or in the initial table of results (table 2) for a better horizontal baseline.
>
> **Hyperparameter settings.** We thank you for pointing out the lack of clarity as to which parameters to use. Since we are working with clinicians and a patients' organization, the goal is to have a well separated and interpretable clustering of the documents. Therefore, we used $\tau=1$ with no OR for "real-life" results that were presented to these publics. Indeed, as seen in table 2, this was the best setting for downstream performance. We could definitely mention this in the discussion.
>
> **Batch Size Sensitivity.** We acknowledge the vulnerability of contrastive learning to batch sampling. We have added a sensitivity analysis for batch sizes \{16,32,64,128,256\} for $\tau=1$ and 0.1. The geometry metrics indicate a low sensitivity of our framework to batch-size. Regarding clustering metrics ($\tau=1, no OR), the sensitivity to batch size is also marginal, but the ARI is degraded (around 0.4) for bigger batch sizes than 32 (0.81). We think that larger batch sizes drastically reduce the number of updates per epoch, leading to poorer convergence.
>
> **Conclusion.** We hope we have addressed the reviewers' concerns by providing a more rigorous validation of the HMV-CL framework through extensive additional experiments and clarifications. By evaluating the model on a large English general health dataset and another specialized French rare disease dataset, we have demonstrated that while standard models perform adequately on common vocabulary, the added value of HMV-CL is most pronounced in highly specialized, low-resource settings. Furthermore, the inclusion of domain-specific baselines and of a batch-size sensitivity analysis confirm that our method is a robust and stable solution to embedding anisotropy.
>
> Naseem, U., Kim, J., Khushi, M., and Dunn, A. G. Identification of disease or symptom terms in reddit to improve health mention classification. In *Proceedings of the ACM Web Conference 2022*, pp. 2573–2581, 2022.

---

> > ### Author Rebuttal · Reviewer_SC8p · 2026-04-01
> >
> > The authors have clearly put significant effort into addressing my initial concerns. I appreciate the evaluation on both the general health Reddit dataset and the additional specialized dataset to demonstrate generalizability. Furthermore, including the domain-specific baselines (CamemBERT-bio and DrBERT), clarifying the recommended hyperparameter settings, and conducting the batch size sensitivity analysis are all valuable additions that help contextualize the model's performance. Thank you as well for committing to fix the presentation issues regarding Figure 1 and Equation 3.
> >
> > While I appreciate the effort to address my concerns, I would require further clarification on the following points to change my score:
> >
> > * Can you provide further analysis supporting the argument that the method is less pronounced on general health datasets because the vocabulary is more common and the data distribution is closer to their pretraining corpora?
> > * Regarding the secondary datasets, could you clarify what specific baseline model was used for the 1,095-message developmental and epileptic encephalopathies dataset, as well some further statistics?
> > * I note the specific values you provided for CamemBERT-bio and DrBERT in the rebuttal. As you suggested, could you please provide the precise downstream task results?
> > * Could you provide the complete tabular results for the batch size ablation study (sizes 16, 32, 64, 128, 256) so the exact degradation across all metrics can be fully assessed?

---

> > > ### Author Response · Authors · 2026-04-07
> > >
> > > **About the method on general health data vs rare disease data**: we are not sure to fully understand the question but will try to answer. Using our method on the general health data from Reddit, we obtained similar downstream performance metrics than with pretrained general embeddings (see tables below). We think this shows our framework generalizes to other languages than French and to data that are not specifically about rare diseases. Here the reddit messages don’t use a very specialized language compared to families confronted to extremely rare diseases and discussing them on a dedicated group, and we hypothesize that RoBERTa was trained on similar data.
> > > To clarify, we would like to share with the reviewer all supervised classification metrics.
> > > - RoBERTa
> > > | |precision|recall|F1|
> > > | ------------- | ------------- | ------------- | ------------- |
> > > |class 0|0.69|0.78|0.73|
> > > |class 1|0.76|0.59|0.66|
> > > |class 2|0.65|0.73|0.69|
> > > |macro avg|0.70|0.70|0.69|
> > >
> > > - Ours ($\tau=1$, no OR)
> > > | |precision|recall|f1|
> > > | ------------- | ------------- | ------------- | ------------- |
> > > |class 0|0.70| 0.76|0.73|
> > > |class 1|0.77| 0.57|0.65|
> > > |class 2| 0.64| 0.78|0.70|
> > > |macro avg|0.71|0.70|0.70|
> > >
> > > **About the 1095 message DEE dataset**: for the embeddings, the baseline used was the same as through the paper, namely CamemBERT-v2, as for the classification task, we only used a simple MLP with AdamW optimizer, trying some different learning rates and number of epochs. We are aware that it is probably not the best classifier, but we thought it was useful to compare our embeddings to baseline embeddings. Here are some further metrics. We also want to specify that the dataset is imbalanced (in the test set 184 documents for the majority class vs 35 documents for the minority class).
> > > - Baseline macro avg: precision=0.42, recall=0.5, f1=0.46
> > > - Baseline weighted avg: precision=0.71, recall=0.84, f1=0.77
> > > - Ours macro avg: precision=0.76, recall=0.71, f1=0.73
> > > - Ours weighted avg: precision=0.86, recall=0.87, f1=0.86
> > >
> > > **Downstream task results for CamemBERT-bio and DrBERT baselines**:
> > >
> > > - Silhouette($\uparrow$): CamemBERT-v2 0.17 - DrBERT 0.14 - CamemBERT-bio 0.11
> > > - Davies-Bouldin($\downarrow$): CamemBERT-v2 1.79 - DrBERT 1.99 - CamemBERT-bio 1.48
> > > - Outliers ratio($\downarrow$): CamemBERT-v2 95.15% - DrBERT 95.66% - CamemBERT-bio 87.12%
> > >
> > > **Complete tabular results for batch size sensitivity study**:
> > >
> > > - Geometry metrics ($\tau=0.1$, no OR)
> > > | Batch size |Uniformity ($\downarrow$)| IC ($\uparrow$) | Avg.cos.sim ($\downarrow$) | Effective Rank (% of total dim)
> > > | ------------- | ------------- | ------------- | ------------- | ------------- |
> > > | 16 | -2.993 | 0.049 | 0.144 | 44 (68.8%)|
> > > | 32 | -3.144 | 0.055 | 0.108 | 45 (70.3%)|
> > > | 64 | -3.193 | 0.053 | 0.090 | 45 (70.3%)|
> > > | 128 | -3.286 | 0.046 | 0.058 | 44 (68.8%)|
> > > | 256 | -3.213 | 0.037 | 0.051 | 42 (65.6%)|
> > > - Geometry metrics ($\tau=1$, no OR)
> > > | Batch size |Uniformity ($\downarrow$)| IC ($\uparrow$) | Avg.cos.sim ($\downarrow$) | Effective Rank (% of total dim)
> > > | ------------- | ------------- | ------------- | ------------- | ------------- |
> > > | 16 | -2.162 | 0.008 | 0.091 | 23 (35.9%)|
> > > | 32 | -2.301 | 0.009 | 0.061 | 25 (39.1%)|
> > > | 64 | -2.367 | 0.010 | 0.050 | 27 (42.2%)|
> > > | 128 | -2.368 | 0.011 | 0.036 | 28 (43.8%)|
> > > | 256 | -2.305 | 0.012 | 0.034 | 30 (46.9%)|
> > > - Downstream metrics ($\tau =1$, no OR, 15 runs)
> > > | Batch size |Silhouette ($\uparrow$)| Davies-Bouldin($\downarrow$) | ARI ($\uparrow$) | Outliers ratio ($\downarrow)
> > > | ------------- | ------------- | ------------- | ------------- | ------------- |
> > > | 32 | 0.241 | 1.034 | 0.806 | 10.78%|
> > > | 64 | 0.249 | 1.066 | 0.829 | 9.82%|
> > > | 128 | 0.224 | 1.120 | 0.903 | 5.81%|
> > > | 256 | 0.231 | 1.179 | 0.870 | 5.20%|
> > >
> > > We apologize, as when we wrote the initial rebuttal, we had to do a lot of complementary experiments in a short time, and we made a mistake in our answer regarding downstream metrics sensitivity to batch size. As you can see from the above tables, our claim that "Regarding clustering metrics ($\tau=1$, no OR), the sensitivity to batch size is also marginal, but the ARI is degraded (around 0.4) for bigger batch sizes than 32 (0.81)" is erroneous. Indeed, the ARI is not degraded for bigger batch sizes, it is better. In the same way, the outliers ratio is better for larger batch sizes. Silhouette and Davies-Bouldin seem to be less sensitive to batch size and are slightly better for smaller batch sizes. We really apologize for the mistake in our initial rebuttal and thank the reviewer for pointing out the importance of batch size.
> > >
> > > **Conclusion**
> > > We thank the reviewer for the really constructive discussion. We are working on a revised version taking into account all the elements of this discussion (changes to fig1, eq3, other domain-specific baselines, two extra corpuses and corresponding downstream tasks, explanation of what parameters to use in the discussion part of the paper, addition of batch size ablation study).

---

### Official Review · Reviewer_xdex · 2026-03-12

**Soundness:** 3
**Presentation:** 2
**Significance:** 2
**Originality:** 2
**Overall Recommendation:** 2
**Confidence:** 4

**Summary:**

This paper introduces a Heterogeneous Multi-View Contrastive Learning (HMV-CL) framework targeting the representation learning challenges of rare disease patient narratives on social media in High Dimensional - Low Sample Size (HDLSS) settings. Within a shared latent space, this framework aligns dense pretrained text embeddings with sparse symbolic views, specifically clinical signs, medical procedures, and pharmacological treatments. Exploiting the low correlation among heterogeneous views as an implicit regularizer, this approach effectively mitigates the embedding anisotropy degradation in pretrained language models without requiring explicit orthogonality penalties, which consequently enhances both the stability and quantitative performance in downstream unsupervised clustering.

**Compliance With Llm Reviewing Policy:**

Affirmed.

**Key Questions For Authors:**

1.This paper establishes only a single-modal text baseline; why were other mainstream contrastive learning frameworks not included for comparative reference?

2.Although the dataset is explicitly stated to be "unlabeled," the paper reports the Adjusted Rand Index (ARI), a metric highly dependent on ground truth labels; please provide the construction methodology for the ground truth used to calculate the ARI in the downstream clustering task, along with its annotation consistency verification process.

3.This paper trains a multi-view model on a limited sample of 784 instances. To address the risk of overfitting inherent in small-sample datasets, it is highly recommended to include a cross-validation stability analysis or provide empirical evidence of generalizability using other publicly available multi-view medical datasets.

4.There is a lack of decoupled analysis regarding the independent contributions of each heterogeneous view; please incorporate view-specific ablation studies to verify the precise contribution of individual modalities in mitigating dimensional collapse.

5.As the validation is restricted to a single-disease French dataset, it is recommended to explore the framework's transferability to English or multilingual rare disease communities, or elaborate on how the model architecture accommodates the heterogeneity involved in cross-lingual medical entity alignment.

**Limitations:**

This paper lacks an adequate discussion regarding its research limitations. The authors merely outlined three superficial limitations—the generalization constraints of a single-disease cohort, the absence of standardized domain benchmarks, and the lack of formal convergence proofs—while failing to address critical flaws in the core experimental design: they neglected to provide horizontal comparisons against mainstream contrastive learning frameworks, elucidate the ground-truth construction logic for the ARI metric given unlabeled data, analyze the overfitting risks associated with small-sample training, and conduct a decoupled analysis of the heterogeneous views' contributions, resulting in a severely inadequate exploration of the method's core limitations and applicability boundaries.

**Strengths And Weaknesses:**

Strengths

Significance: By focusing on representation learning for rare disease social media narratives in High Dimensional - Low Sample Size (HDLSS) settings, the paper presents a clear motivation and holds practical relevance for real-world public health text mining.

Originality: The proposed heterogeneous multi-view contrastive learning framework empirically demonstrates the natural regularization effect of heterogeneous views, effectively circumventing the representation dimensional collapse issue without requiring explicit constraints.

Soundness: A dual-dimensional evaluation system comprising embedding space geometry and downstream clustering is constructed, with robustness ensured by multiple randomized runs; furthermore, multidimensional objective metrics—including uniformity, isotropic coefficient, and effective rank—are utilized to validate the framework's capability to mitigate the embedding anisotropy degradation inherent in pretrained language models.

Weaknesses

1.The experimental design suffers from a critical lack of horizontal baselines, as the ablations and evaluations in this paper are exclusively compared against a unimodal text baseline.

2.Although the paper explicitly states the use of 784 unlabeled social media posts, it reports the Adjusted Rand Index (ARI)—a metric highly dependent on Ground Truth labels—in downstream tasks, yet the logic for constructing these ground truth labels appears to be missing.

3.The validation of the method's generalizability is notably inadequate, having only been tested on a single-disease, small-sample French dataset without cross-scenario evaluations, which fails to support claims of the framework's generalizability as a universal representation learning method.

4.The model validation entails a high risk of overfitting in small-sample scenarios; specifically, training a multi-view network under the low-resource setting of merely 784 samples from a single French rare disease cohort poses a significant small-sample overfitting risk, yet no explicit mechanisms for overfitting prevention are provided.

---

> ### Author Rebuttal · Authors · 2026-03-30
>
> We would like to sincerely thank the reviewer for the rigorous analysis of our experimental design and the insightful comments.
>
> **Horizontal baselines.** To address the lack of horizontal baselines, we have expanded our comparative study. We include the two domain-specific pre-trained models mentioned in the introduction of the paper (DrBERT and CamemBERT-bio). In both cases, uniformity is $\infty$ like for CamemBERT-v2 and IC is in the same order of magnitude ($10^{-9}$). Regarding effective rank, CamemBERT-bio utilizes 23\% of the available space (close to the 27\% of CamemBERT-v2) while DrBERT utilizes close to 40\%. Downstream task performance metrics are in the same range as our initial baseline. All these figures can be included as an extra appendix or in the initial table of results (table 2) for a better horizontal baseline.
>
> Regarding other methods addressing embedding anisotropy, we are currently in the process of adding, Bert-Flow and SimCSE as baselines. However, the results are not yet ready at the time of rebuttal.
>
> **Clarification of ARI.** We apologize for the confusion regarding the ARI. In our unsupervised setting, ARI is used as a measure of clustering stability (inter-run consistency) rather than accuracy against a ground truth. We calculated the ARI pairwise between 15 independent randomized runs.
>
> **Generalizability and overfitting risk.** We understand the concerns raised regarding the small size of the sample and have incorporated several design choices to mitigate that risk. The use of the ARI to evaluate the stability of our downstream clustering task indicates that the resulting latent spaces are not a product of noise but are indeed stable and reproducible structures.
>
> Regarding generalizability, we have conducted a second experiment on the public dataset from Naseem et al. (2022)'s paper*. This dataset consists of 10,015 Reddit posts written in English and coming from subreddits where users are likely to discuss topics related to health. Our method (tested with $\tau=1$/no OR) improves geometry metrics compared to the baseline (roberta-base). Indeed, we report a decrease of mean cosine similarity (0.01 compared to 0.95), of IC ($1.4 \times 10^{-3}$ compared to $8.6 \times 10^{-8}$) and of uniformity (-3.37 compared to $\infty$).
>
> The downstream task associated to this dataset is the classification of these posts into three balanced categories (personal health mentions, non-personal and figurative/hyperbolic). We used a simple MLP trained using AdamW optimizer to evaluate the discriminative capacity of our embeddings (train/test split of 0.8/0.2). We achieved similar performance with the baseline embeddings and ours, with macro F1 of 0.69 and 0.70 respectively.
>
> We did the same experiment (same simple architecture, same train/test split, same optimizer) on a classification task on another dataset consisting of 1,095 labeled (by two annotators) messages mentioning various developmental and epileptic encephalopathies on social media. The objective was to differentiate mentions of personal experience of the disease and other discourses (imbalanced dataset). In this HDLSS context, our embeddings ($\tau=1$/no OR) significantly outperformed the baseline, increasing the weighted-average F1 from 0.77 to 0.86.
>
> These additional results indicate that while standard pretrained models perform adequately on "general health" social media posts (likely because the vocabulary is more common and the data distribution is closer to their pretraining corpora), the added value of HMV-CL is most pronounced in highly specialized domains.
>
> **View-specific ablation studies.** We performed a view-specific ablation analysis (for $\tau = 1$ and 0.1) which demonstrates that the addition of even a single symbolic view is sufficient to improve latent space geometry, reducing the mean cosine similarity from 0.88 to less than 0.08 and the uniformity from $\infty$ to values between -2 and -4 for every possible combination. The increase in effective rank of the full model compared to partial view combinations (from around 15 to 26 for $\tau=1$ and from around 30 to 45 for $\tau=0.1$) demonstrates that combining all views is essential to achieving the maximum expressive capacity of the 64-dimensional shared latent space.
>
> **Conclusion.** We hope we have addressed the reviewers' concerns by providing a more rigorous validation of our framework through our additional experiments and clarifications.
>
> Naseem, U., Kim, J., Khushi, M., and Dunn, A. G. Identification of disease or symptom terms in reddit to improve health mention classification. In *Proceedings of the ACM Web Conference 2022*, pp. 2573–2581, 2022.

---

> > ### Author Rebuttal · Reviewer_xdex · 2026-04-03
> >
> > Thanks for the detailed rebuttal. The ARI clarification and the new view-specific ablations definitely help address my initial points. That being said, given the extremely small sample size in this HDLSS setting, there remains a risk of overfitting on the dataset.

---

> > > ### Author Response · Authors · 2026-04-07
> > >
> > > We thank the reviewer for acknowledging our rebuttal and for raising the important concern about overfitting, that we missed replying to in our first rebuttal.
> > >
> > > In this work the primary goal is to learn a representation that best captures the structure of this specific dataset, rather than achieving strong out‑of‑domain generalization. In that sense, a certain degree of fitting to the dataset is acceptable, as long as the improvement is consistent across multiple geometry and downstream metrics, and on other datasets/tasks too.
> > >
> > > To mitigate the overfitting risk, we kept the model architecture simple and used fixed hyperparameters across datasets.
> > >
> > > We explicitly acknowledge the overfitting risk as a limitation in the revised manuscript and highlight that the results should be interpreted in the context of the specific HDLSS regime considered. We also clarify the ARI and add the view-specific ablations.

---

### Official Review · Reviewer_oDq3 · 2026-03-13

**Soundness:** 2
**Presentation:** 3
**Significance:** 2
**Originality:** 2
**Overall Recommendation:** 3
**Confidence:** 3

**Summary:**

This paper proposes the HMV-CL framework, a heterogeneous multi-view contrastive learning model for learning document-level representations of rare disease patient narratives from French social media data. It tackles one main challenges: How to combine text embeddings to solve the embedding anisotropy issue. The experimental results demonstrate the effectiveness of the proposed model.

**Compliance With Llm Reviewing Policy:**

Affirmed.

**Final Justification:**

The authors have provided some new results during rebuttal, but the gaps in experimental scope prevent me from upgrading my score. I will maintain my current evaluation but slightly lower my confidence, and I would not strongly oppose a consensus-based decision if the other reviewers lean toward acceptance.

**Key Questions For Authors:**

1. If terminology heterogeneity is the main challenge, why are medical LLMs or stronger biomedical language models not included as baselines or discussion points?

2. Can the authors clarify what is algorithmically novel here beyond assembling standard multi-view contrastive components?

3. Is there evidence that the method generalizes beyond this specific disease/domain setting?

**Limitations:**

See weakness part.

**Strengths And Weaknesses:**

Strength:

1.The paper studies a practically relevant problem in the medical domain, where narratives from rare disease patients and families on social media can offer valuable insights for public health researchers.

2.The paper is well-motivated by the challenges of anisotropy and heterogeneous domain-specific medical language in rare disease narratives, which is a topic worth exploring.

3.The paper provides a well-structured overview, making it easy to follow for readers.

4.Experimental results demonstrate the effectiveness of the proposed method.

Weakness:

1.About novelty: The technical novelty is limited. The method appears to consist of standard component in multiview-learning: pretrained text embeddings, view-specific encoders, contrastive alignment, and final concatenation. While the application is interesting, the algorithmic core does not appear substantially new. At present, the work reads more like an adaptation of existing representation learning techniques to a new biomedical dataset than a genuine methodological advance.

2.About Significance: The authors repeatedly emphasize that narratives are highly heterogeneous, mixing highly specialized and potentially rare medical vocabulary and informal and experiential discourses. However, this type of challenge increasingly appears to be better addressed by modern medical LLMs, or even strong general-purpose LLMs, which are generally more capable than BERT-based models at recognizing professional medical terms and aligning semantically heterogeneous expressions. As a result, it is unclear whether the proposed framework represents the most meaningful or future-proof solution to the problem, which weakens the overall significance of the contribution.

3.About Soundness: While the overall pipeline appears technically plausible, I don't understand why the proposed HMV-CL framework can work for the narratives. From the current presentation, the approach seems to rely largely on standard multi-view learning and contrastive learning components, both of which have been extensively studied in prior works. However, the paper does not clearly explain why this particular combination should be especially effective for the targeted heterogeneity problem. As a result, although the method may work empirically, its technical justification remains somewhat underdeveloped.

4.About Experiments: It would also be helpful if the authors could provide more interpretable intermediate outputs to better illustrate how the model makes decisions. For example, since the framework relies on heterogeneous views and contrastive alignment, the paper could present examples showing how specific clinical signals influence the learned embeddings or provided the textual outputs.

---

> ### Author Rebuttal · Authors · 2026-03-30
>
> We thank the reviewer for recognizing the practical relevance of our work and the interest of the topic.
>
> **Domain-specific LMs and LLMs.** We added two domain-specific pretrained models (CamemBERT-bio and DrBERT, mentioned in the introduction of the paper) as alternative baselines. In both cases, uniformity is $\infty$ like for CamemBERT-v2 and IC is in the same order of magnitude ($10^{-9}$). Regarding effective rank, CamemBERT-bio utilizes 23\% of the available space (close to the 27\% of CamemBERT-v2) while DrBERT utilizes close to 40\%. Downstream task performance metrics are in the same range as our initial baseline. All these figures can be included as an extra appendix or in the initial table of results (table 2) for a better horizontal baseline.
> While LLMs are powerful, our framework offers a sovereign, private, and low-resource solution essential for small patients' organizations with sensitive data. We recognize that these solutions should however be discussed in the discussion of the paper.
>
> **Clarification of the methodological novelty.** While we acknowledge that multi-view contrastive learning is an established representation learning technique, we would like to point out the contributions of our paper. Firstly, unlike standard multi-view setups that often use data augmentation or parallel corpora, our framework aligns with the pretrained text embeddings sparse symbolic views, extracted directly from the patients' narratives. The other algorithmical novelty is that the heterogeneity of these views act as a natural regularizer. Indeed, we showed that the low inter-view correlations are enough to prevent dimensional collapse, eliminating the need for explicit penalties often required in other contrastive learning frameworks.
>
> **Generalizability to other domains.** We have conducted a second experiment the public dataset from Naseem et al. (2022)'s paper*. This dataset consists of 10,015 Reddit posts written in English and coming from subreddits where users are likely to discuss topics related to health. Our method ($\tau=1$/no OR) improves geometry metrics compared to the baseline (roberta-base): mean cosine similarity (0.01 compared to 0.95), IC ($1.4 \times 10^{-3}$ compared to $8.6 \times 10^{-8}$) and uniformity (-3.37 compared to $\infty$).
>
> The downstream task associated to this dataset is the classification of these posts into three balanced categories (personal health mentions, non-personal and figurative/hyperbolic). We used a simple MLP trained using AdamW optimizer to evaluate the discriminative capacity of our embeddings (train/test split of 0.8/0.2). We achieved similar performance with the baseline embeddings and ours, with macro F1 of 0.69 and 0.70 respectively.
>
> We did the same experiment (same simple architecture, same train/test split, same optimizer) on a classification task on another dataset consisting of 1,095 labeled (by two annotators) messages mentioning various developmental and epileptic encephalopathies on social media. The objective was to differentiate mentions of personal experience of the disease and other discourses (imbalanced dataset). In this HDLSS context, our embeddings ($\tau=1$/no OR) significantly outperformed the baseline on downstream classification task, increasing the weighted-average F1 from 0.77 to 0.86.
>
> These additional results indicate that while standard pretrained models perform adequately on "general health" social media posts (likely because the vocabulary is more common and the data distribution is closer to their pretraining corpora), the added value of HMV-CL is most pronounced in highly specialized domains.
>
> **View-specific ablation study.** We would like to point out to our rebuttal to reviewer xdex, where we explain the view ablation study, and hope this ablation study can satisfy the reviewer regarding how each view influences the learned embeddings.
>
> **Conclusion.** We hope we have adressed the reviewer's concerns  with the additional empirical evidence in this rebuttal. The integration of strong biomedical baselines confirms that our multi-view approach offers a unique advantage for handling the heterogeneity of patients' narratives in specialized settings. Our tests demonstrate the generalizability of our method, although it seems to have more importance in HDLSS and rare disease contexts. We have also clarified the algorithmic novelty of our framework.
>
> Naseem, U., Kim, J., Khushi, M., and Dunn, A. G. Identification of disease or symptom terms in reddit to improve health mention classification. In *Proceedings of the ACM Web Conference 2022*, pp. 2573–2581, 2022.

---

> > ### Author Rebuttal · Reviewer_oDq3 · 2026-04-04
> >
> > Thank you for the rebuttal. However, the added baselines (CamemBERT-bio and DrBERT) are still BERT-scale models and do not address the original concern of why some open LLMs were not included as baselines. I will maintain my current score.

---

> > > ### Author Response · Authors · 2026-04-07
> > >
> > > We thank the reviewer for acknowledging our rebuttal and apologize about not addressing the concern about open LLMs in our initial rebuttal, as we did not have enough time before the initial deadline.
> > >
> > > We would like to share with the reviewer our results, obtained by using the instruction‑tuned embedding model *intfloat/e5‑mistral‑7b‑instruct* (Wang et al., 2023) and prefixing each document with a generic task in the format Instruct: $<$task$>$\nDocument: $<$text$>$. Document embeddings are obtained via last‑token pooling and L2‑normalization, following the E5‑Mistral recommendation.
> > >
> > > **Results**
> > > - Geometry metrics
> > > | Batch size |Uniformity ($\downarrow$)| IC ($\uparrow$) | Avg.cos.sim ($\downarrow$) | Effective Rank (% of total dim)
> > > | ------------- | ------------- | ------------- | ------------- | ------------- |
> > > | CamemBERT-v2 | $\infty$ | 8.25 $\times 10^{-9}$  | 0.880 | 209 (27.21%)|
> > > | LLM | -0.672 | 4.44 $\times 10^{-16}$ | 0.829 | 368 (8.98%)|
> > > | Ours ($\tau=1$, no OR) | -2.39 | 0.009 | 0.05 | 26 (40.6%)|
> > > | Ours ($\tau=0.1$, no OR) | -3.17 | 0.053 | 0.10 | 45 (70.3%)|
> > >
> > > - Downstream metrics
> > > | Batch size |Silhouette ($\uparrow$)| Davies-Bouldin($\downarrow$) | Outliers ratio ($\downarrow)
> > > | ------------- | ------------- | ------------- | ------------- |
> > > | CamemBERT-v2 | 0.17 | 1.79 | 95.1%|
> > > | LLM | 0.05 | 1.80 | 50.0 %|
> > > | Ours ($\tau=1$, no OR) | 0.25 | 1.04| 10.6 %|
> > >
> > > **Discussion**: regarding geometric characteristics, the LLM embeddings present a better uniformity compared to plain camemBERT-v2 embeddings. However, the effective rank is much lower, meaning that a lot of the 4096 dimensions are unuseful.
> > >
> > > Regarding downstream performance, Silhouette and Davies-Bouldin are close to camemBERT-v2 embeddings, while the proportion of documents unclassified is lower (50% vs 95% for camemBERT-v2 embeddings). However, on all metrics considered, our framework does better than LLM embeddings.
> > >
> > > This was a relatively light weight LLM, so maybe some bigger LLMs could do better. However, given the computation time (28 minutes for our small dataset compared to 30 sec for the computation of camemBERT-v2 embeddings) and the negative externalities (regarding CO2 emissions, water consumption,…), we don’t think LLM embeddings are a solution to our issues.
> > >
> > > **Conclusion**:  we showed the reviewer that our framework could generalize to English, and to social media data about “general health”, and not only to the rare disease narratives case. We also performed a complementary experiment based on an open source LLM. These embeddings do slightly better on geometry metrics and downstream proportion of uncategorized documents than plain pretrained BERT embeddings, but our embeddings improve all metrics compared to LLM embeddings. We will mention these complementary results in our revised manuscript, as well as the ones presented in the initial rebuttal and we will make sure to discuss the use of LLMs in the discussion.

---

### Official Review · Reviewer_92u6 · 2026-03-13

**Soundness:** 2
**Presentation:** 3
**Significance:** 2
**Originality:** 3
**Overall Recommendation:** 4
**Confidence:** 3

**Summary:**

Pretrained text embeddings usually suffer from embedding anisotropy: they are concentrated in a few dominant directions. This phenomenon is correlated with poor downstream tasks. In domains where the data distribution differs greatly from the pretrained model, these effects are stronger. The paper presents a method for training less anisotropic document-level embeddings in the context of French rare disease social media data. They present experimental results comparing the embedding space geometry of their method against that of a pretrained model and compare performance on clustering tasks.

**Compliance With Llm Reviewing Policy:**

Affirmed.

**Final Justification:**

The SimCSE and BERT-Flow comparisons directly address my remaining concern. The results show clear advantages over these alternatives, particularly in clustering quality and outlier reduction. I am updating my score to a weak accept.

**Key Questions For Authors:**

1) What reference clustering is used to compute the Adjusted Rand Index (ARI)? The paper does not specify what ground truth the BERTopic output is compared against. Clarifying this is essential to interpret the reported ARI values.
2) Have you compared HMV-CL against other methods that address embedding anisotropy? If so, how does the method compare? This would help assess whether the multi-view approach offers advantages over simpler alternatives.
3) Have you evaluated the embeddings on downstream tasks beyond unsupervised clustering, such as semantic textual similarity or classification? Results on additional tasks would strengthen the claim that HMV-CL produces generally useful representations.
4) Has the method been evaluated on datasets beyond the current rare disease corpus?
Evidence of generalization to other domains or languages would help clarify whether the contribution is primarily methodological or domain-specific.

**Limitations:**

Yes

**Strengths And Weaknesses:**

Strengths
The paper is generally well written and clearly presented, though some methodological details are underspecified (e.g., handling of missing views, reference labels for ARI).
The paper identifies a relevant and well-documented problem. The proposed method combines known components (contrastive loss, multi-view learning) in a reasonable way, with the main conceptual contribution being the argument that heterogeneous views act as natural regularizers against dimensional collapse.
Based on embedding space geometry metrics, the experiments show that the method reduces anisotropy compared to naive pretrained embeddings.
The paper presents appropriate ablation studies that help understand the individual contributions and effects of the method's key hyperparameters (temperature τ and orthogonality regularization weight λ).

Weaknesses
There is no comparison with other baseline methods. The only reference point is raw CamemBERT-v2 mean-pooled embeddings, with no adaptation. Natural baselines are absent, such as BERT-flow (Li et al., 2020), which addresses the same anisotropy problem by transforming embeddings into an isotropic Gaussian distribution without any training, or contrastive sentence embedding methods like SimCSE (Gao et al., 2021).
The evaluation of downstream tasks is limited to unsupervised clustering, which is insufficient to assess the embeddings' utility more broadly. Relevant tasks such as semantic textual similarity or classification (e.g., categorizing posts by topic) are not considered.
The contrastive learning method appears to be more generally applicable beyond rare disease social media data. The current application reads as a suitable case study, yet  it is presented as a central part of the contribution, which hinders the paper's significance. The paper would benefit from evaluating the method on additional datasets to assess its generality

Minor comments:
The Adjusted Rand Index (ARI) requires reference labels to compare against, yet the paper does not specify what ground truth clustering is used. Given that the task is presented as unsupervised and the dataset is described as unlabeled, it is unclear how ARI is computed and how the reported values should be interpreted.

---

> ### Author Rebuttal · Authors · 2026-03-30
>
> We would like to thank the reviewer for pointing out the strengths of our paper and for the precise comments on baseline methods and the insightful request for a broader evaluation of our downstream tasks.
>
> **Clarification of ARI.** We apologize for the confusion regarding the ARI. In our unsupervised setting, ARI is utilized as a measure of clustering stability (inter-run consistency) rather than accuracy against a ground truth. We calculated the ARI pairwise between 15 independent randomized runs.
>
> **Missing Baselines.** To better assess the advantages of our multi-view approach over simpler alternatives, we have taken the following actions.
> (i) We added two pretrained domain-specific models (CamemBERT-bio and DrBERT, mentioned in the introduction of the paper) as alternative baselines. In both cases, uniformity is $\infty$ like for CamemBERT-v2 and IC is in the same order of magnitude ($10^{-9}$). Regarding effective rank, CamemBERT-bio utilizes 23\% of the available space (close to the 27\% of CamemBERT-v2) while DrBERT utilizes close to 40\%. Downstream task performance metrics are in the same range as our initial baseline. All these figures can be included as an extra appendix or in the initial table of results (table 2) for a better horizontal baseline.
> (ii) Regarding other methods addressing embedding anisotropy, we are currently in the process of adding, as suggested by the reviewer, Bert-Flow and SimCSE as baselines. However, the results are not yet ready at the time of rebuttal.
>
> **Additional downstream task and datasets.** We have conducted a second experiment the public dataset from Naseem et al. (2022)'s paper*. This dataset consists of 10,015 Reddit posts written in English and coming from subreddits where users are likely to discuss topics related to health. Our method ($\tau=1$/no OR) improves geometry metrics compared to the baseline (roberta-base): mean cosine similarity (0.01 compared to 0.95), IC ($1.4 \times 10^{-3}$ compared to $8.6 \times 10^{-8}$) and uniformity (-3.37 compared to $\infty$).
>
> The downstream task associated to this dataset is the classification of these posts into three balanced categories (personal health mentions, non-personal and figurative/hyperbolic). We used a simple MLP trained using AdamW optimizer to evaluate the discriminative capacity of our embeddings (train/test split of 0.8/0.2). We achieved similar performance with the baseline embeddings and ours, with macro F1 of 0.69 and 0.70 respectively.
>
> We did the same experiment (same simple architecture, same train/test split, same optimizer) on a classification task on another dataset consisting of 1,095 labeled (by two annotators) messages mentioning various developmental and epileptic encephalopathies on social media. The objective was to differentiate mentions of personal experience of the disease and other discourses (imbalanced dataset). In this HDLSS context, our embeddings ($\tau=1$/no OR) significantly outperformed the baseline, increasing the weighted-average F1 from 0.77 to 0.86.
>
> These additional results indicate that while standard pretrained models perform adequately on "general health" social media posts (likely because the vocabulary is more common and the data distribution is closer to their pretraining corpora), the added value of HMV-CL is most pronounced in highly specialized domains. They also show that HMV-CL produces useful representations for less specialized data (the English corpus being about health "in general"). Moreover, the experiment on Reddit data shows that the framework is adaptable to other languages than French. Finally, they bring to the paper another type of downstream task (supervised classification) that was lacking from the first version, as pointed out by the reviewer.
>
> **Conclusion.** We hope we have adressed most of the reviewers' concerns by adding baselines and two supervised classification tasks, including one on a more general dataset, written in English.
>
> Naseem, U., Kim, J., Khushi, M., and Dunn, A. G. Identification of disease or symptom terms in reddit to improve health mention classification. In *Proceedings of the ACM Web Conference 2022*, pp. 2573–2581, 2022.

---

> > ### Author Rebuttal · Reviewer_92u6 · 2026-04-04
> >
> > Thank you for the clarifications on ARI and for including experiments on the Reddit and encephalopathy datasets. It would be helpful to make the explanation of ARI as a measure of clustering stability more explicit in the paper.
> >
> > The new classification results are appreciated and address my concerns about non-clustering tasks. The Reddit/encephalopathy comparison supports the paper’s claims of strength under specialized domains.
> >
> > The missing anisotropy baselines are not trivial. The reported metrics and downstream performance appear similar to the original CamemBERT baselines. Further comparison with BERT-Flow and SimCSE would help the reader assess whether the multi-view contrastive approach offers a concrete advantage over simpler alternatives, like fine-tuning.

---

> > > ### Author Response · Authors · 2026-04-07
> > >
> > > We thank the reviewer for the rebuttal acknowledgment and will make sure to make the explanation of ARI as a measure of clustering stability clearer in the revised manuscript.
> > >
> > > We also thank the reviewer for the suggestion of comparing our framework to other methods than pertained BERT models. We followed the suggestion and compared SimCSE method and BERT-flow method to enrich our baselines, but it took longer than the initial rebuttal date.
> > >
> > > **Methodology**:
> > > For "SimCSE", we fine‑tuned the CamemBERT‑v2 encoder following Gao et al.'s SimCSE framework, using self‑augmented document views: each document is paired with a slightly corrupted version of itself (via random truncation), and the model was trained to pull these two views closer in the embedding space while pushing away other documents in the same batch.
> > >
> > > For BERT‑Flow (Li et al. (2020)), we applied a CamemBERT‑v2‑adapted version of the normalizing‑flow approach: we centered the embeddings and learned a small residual neural network that maps them toward a more isotropic, Gaussian‑like distribution.
> > >
> > > **Results**:
> > > - Geometry metrics
> > > | Batch size |Uniformity ($\downarrow$)| IC ($\uparrow$) | Avg.cos.sim ($\downarrow$) | Effective Rank (% of total dim)
> > > | ------------- | ------------- | ------------- | ------------- | ------------- |
> > > | CamemBERT-v2 | $\infty$ | 8.25 $\times 10^{-9}$  | 0.880 | 209 (27.21%)|
> > > | SimCSE | -0.115| 8.10 $\times 10^{-6}$ | 0.971 | 99 (12.89%)|
> > > | CamemBERT-flow | $\infty$| 3.28$\times 10^{-17}$ | -0.002 | 202 (19.73%)|
> > > | Ours ($\tau=1$, no OR) | -2.39 | 0.009 | 0.05 | 26 (40.6%)|
> > > | Ours ($\tau=0.1$, no OR) | -3.17 | 0.053 | 0.10 | 45 (70.3%)|
> > >
> > > - Downstream metrics
> > > | Batch size |Silhouette ($\uparrow$)| Davies-Bouldin($\downarrow$) | Outliers ratio ($\downarrow)
> > > | ------------- | ------------- | ------------- | ------------- |
> > > | CamemBERT-v2 | 0.17 | 1.79 | 95.1%|
> > > | SimCSE | 0.18 | 1.39 | 68.6 %|
> > > | CamemBERT-flow | -0.11 | 1.45 | 36.5%|
> > > | Ours ($\tau=1$, no OR) | 0.25 | 1.04| 10.6 %|
> > >
> > > **Discussion**: SimCSE significantly improves IC and Uniformity compared to CamemBERT‑v2, while using a smaller proportion of the available dimensions (13% compared to 27%). Compared to CamemBERT-v2 embeddings, this method reduces the proportion of outliers with similar clustering quality (silhouette and Davies-Bouldin).
> > >
> > > CamemBERT‑flow does not improve that much the geometry metrics and has a high cost in informativeness and clustering performance.
> > >
> > > Our framework maintains low uniformity, high informativeness, and a favorable effective rank, while achieving better clustering (higher Silhouette, lower DB, and much lower outliers ratio) than these other methods.
> > >
> > > **Conclusion**: Together with the additional experiments on Reddit and encephalopathy datasets, these results demonstrate that our approach offers a concrete advantage over simpler alternatives, especially in specialized domains such as rare disease narratives on specialized social media.
> > > We will make sure to include these complementary results in the revised manuscript, as well as the clarification of ARI as a stability measure, the domain specific pertained BERT baselines, and the experiments on the other datasets and on supervised classification.

---

### Decision · Program_Chairs · 2026-04-30

**Decision:**

Reject

**Comment:**

The reviewers seem to be in consensus that this paper lies somewhat below the acceptance threshold for ICML, also after taking into consideration the detailed rebuttals submitted by the authors.
I therefore recommend rejection.